# Experience-Guided Reflective Co-Evolution of Prompts and Heuristics for Automatic Algorithm Design

## Abstract

Combinatorial optimization problems are traditionally tackled with handcrafted heuristic algorithms, which demand extensive domain expertise and significant implementation effort. Recent progress has highlighted the potential of automatic heuristics design powered by large language models (LLMs), enabling the automatic generation and refinement of heuristics. These approaches typically maintain a population of heuristics and employ LLMs as mutation operators to evolve them across generations. While effective, such methods often risk stagnating in local optima. To address this issue, we propose the Experience-Guided Reflective Co-**Evo**lution of **P**rompt and **H**euristics (**EvoPH**) for automatic algorithm design, a novel framework that integrates the island migration model with the elites selection algorithm to simulate diverse heuristics populations. In EvoPH, prompts are co-evolved with heuristic algorithms, guided by performance feedback. We evaluate our framework on two problems, i.e., Traveling Salesman Problem and Bin Packing Problem. Experimental results demonstrate that EvoPH achieves the lowest relative error against optimal solutions across both datasets, advancing the field of automatic algorithm design with LLMs.

## 1 Introduction

Combinatorial optimization problems (COPs) (Dantzig & Ramser, 1959) form a fundamental branch of mathematical research. They drive progress in areas such as algorithm design and computational complexity theory, while also providing essential methods for addressing real-world challenges in resource allocation and decision-making. Traditionally, solving COPs relied on handcrafted heuristic algorithms, which require researchers to possess substantial domain knowledge (Pillay & Qu, 2018). Moreover, practical applications often demand customized algorithms with distinct processes and parameters, resulting in considerable human effort (Hromkovič, 2013). To alleviate these challenges, researchers have proposed the paradigm of automatic heuristics design (AHD), with Genetic Programming (GP) being one of the most representative examples (Langdon & Poli, 2013). GP iteratively refines heuristics by applying mutation operators (Duflo et al., 2019). However, the effectiveness of GP-based methods is fundamentally constrained by the human-defined operator set, which not only increases implementation difficulty but also limits achievable performance.

In recent years, large language models (LLMs) have demonstrated remarkable effectiveness across diverse domains, notably through prompt engineering that simulates mutation operations, enabling applications in code generation, automated machine learning, scientific discovery, and algorithm design (Zhao et al., 2023; Jiang et al., 2024; Liu et al., 2024b). Nevertheless, current practices often rely on ineffective evolutionary algorithms or fixed prompts, which limit adaptability in complex scenarios. As a consequence, existing approaches tend to converge prematurely to local optima, while syntax or logic errors introduced during code execution frequently propagate across descendant heuristics, leading to repeated failures and substantial computational overhead.

To address these limitations, we propose **EvoPH**, a novel experience-guided reflective co-**Evo**lution framework that can co-evolve **P**rompts and **H**euristics for automatic algorithm design. EvoPH is built upon an iterative cycle of heuristics generation, evaluation, experience storage, and reflection. In each iteration, new heuristics are generated by an LLM, followed by assessing their performance

through execution, and the outcomes are distilled into experience that informs subsequent heuristic search. During heuristics evolution, the saved experience guides the LLM to evolve heuristics through a diverse set of mutation operators. Specifically, we propose an *island-based elites selection algorithm*, which can preserve diversity while enabling the exchange of high-quality elites across populations. Here, an *island* refers to an independent sub-population that evolves in parallel, occasionally exchanging high-quality elites with others. The core of our EvoPH framework lies in the integration of prompt evolution, where prompts are not only adaptively rewritten but also progressively specialized based on fine-grained execution feedback. This mechanism enables dynamic error correction and knowledge consolidation, allowing prompts to evolve into increasingly task-specific guides that retain effective instructions while continuously steering the evolution of heuristics. Furthermore, we propose an *experience-driven strategy sampling* that selects or combines mutation operators and interacts with prompt evolution to ensure prompts and strategies co-adapt in a self-correcting manner. Through this synergy, prompts function as both adaptive controllers and knowledge carriers, aligning task descriptions with heuristics evolution.

To evaluate the performance of EvoPH, we conduct experiments on the Traveling Salesman Problem (TSP) and the Bin Packing Problem (BPP). We used Gurobi or OR-Tools to calculate the optimal solution and relative error to ensure the authority of the evaluation results. Our core contributions can be summarized as follows:

- We propose EvoPH, an automatic algorithm design framework that co-evolves prompts and heuristics. By iteratively evolving prompts based on execution feedback, leveraging stored experience to inform mutation operator choice, and dynamically selecting evolution strategies, EvoPH generates targeted yet diverse prompts. This synergy enables heuristic algorithms to escape local optima during evolution while promptly correcting errors in code execution.

- We construct benchmark datasets for TSP and BPP by adapting TSPlib (Reinelt, 1991) and BPPlib (Delorme et al., 2018), converting them into distance-matrix formats for efficient evaluation. We also adopt Gurobi or OR-Tools as references for optimal solutions and release all data to facilitate future research in automatic heuristic design.

- Experimental results demonstrate that on TSP, EvoPH significantly improves performance compared to prior frameworks. On BPP, EvoPH effectively enhances baseline heuristics, whereas existing frameworks only yield marginal improvements.

## 2 RELATED WORK

**Neural combinatorial optimization**. Neural combinatorial optimization (NCO) has emerged as a promising paradigm for solving combinatorial optimization problems (COPs) in an end-to-end manner (Chen et al., 2023; Ma et al., 2023). As a variant of hyper-heuristics (HH), it explores heuristic spaces through neural architectures and training algorithms (Romera-Paredes et al., 2024; Liu et al., 2023a). Existing methods are typically grouped into learning constructive heuristics (LCH), which incrementally build solutions (Liu et al., 2023b; Son et al., 2025); learning improvement heuristics (LIH) (André & Kevin, 2020), which refine existing solutions through neural-guided search and hybrid solvers, which combine neural models with classical algorithms (Gasse et al., 2019; Luo et al., 2023). Applications now span routing, SAT, scheduling, and other NP-hard problems (Li et al., 2023; Sun & Yang, 2023), though challenges remain in scalability, generalization, and closing the gap with state-of-the-art classical solvers (Selsam, 2019).

**LLM for Evolutionary computation**. Evolutionary computation (EC) is a population-based black-box optimization paradigm well suited for non-convex or discrete problems without gradient information (Eiben & Smith, 2015; Bäck et al., 1997). With the rise of LLMs, recent work explores their integration with EC frameworks (Chauhan et al., 2025). For instance, the LMEA framework uses natural language instructions to guide LLM-based crossover and mutation on textual solution representations (Liu et al., 2024c), while EvoLLM leverages LLMs in a zero-shot manner to execute full evolutionary cycles via ranking-based prompting, achieving strong results on synthetic benchmarks (Lange et al., 2024). These approaches highlight LLMs as intelligent operators or high-level controllers, showing potential in heuristics design, code generation, and planning.

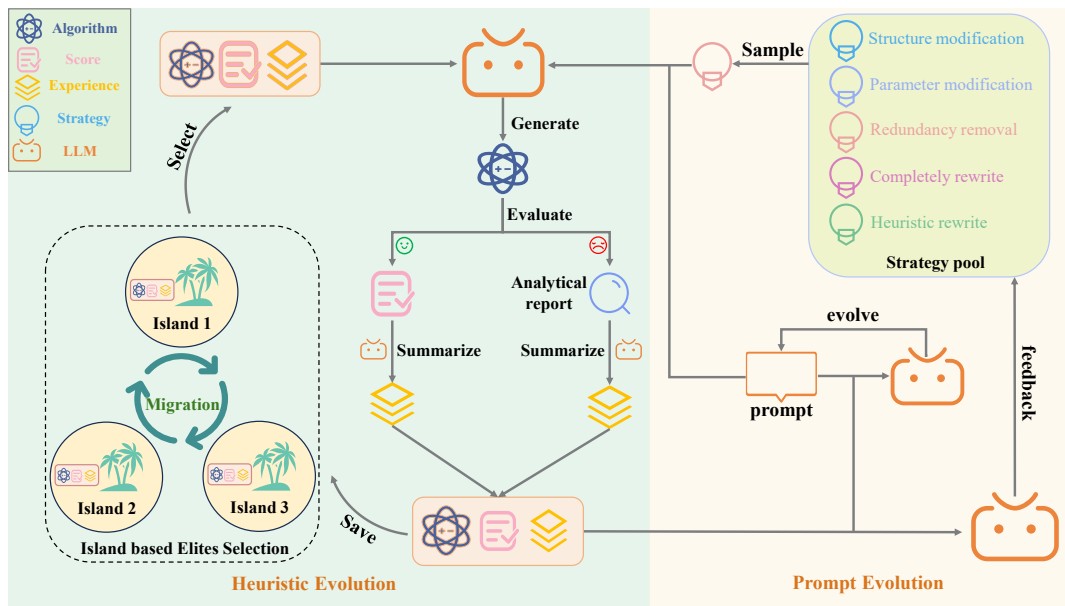

Figure 1: EvoPH comprises two interacting processes. Heuristics Evolution generates, evaluates, and stores candidate algorithms, providing feedback for further search. Prompt Evolution adaptively refines LLM prompts and strategy selection based on this feedback.

## 3 PRELIMINARIES

In this section, We start by presenting two representative NP-hard problems, the Traveling Salesman Problem and the Bin Packing Problem, which serve as running examples throughout this work.

### 3.1 TRAVELING SALESMAN PROBLEM AND BIN PACKING PROBLEM

**Traveling Salesman Problem**. The TSP is a fundamental NP-hard problem in combinatorial optimization. It is defined on a fully weighted graph $G = (V, E)$, where $V = \{v_1, v_2, ..., v_n\}$ is a set of $n$ vertices, and $E$ is the set of edges containing all unordered pairs of distinct vertices, i.e., $E = \{\{v_i, v_j\} \mid v_i, v_j \in V, i \neq j\}$. A weight function $w : E \to \mathbb{R}^+$ assigns a non-negative cost $w(v_i, v_j)$ to each edge. The goal is to find a Hamiltonian cycle, i.e., a cycle that visits each vertex in $V$ exactly once, with the minimum total weight. If a tour is represented by a permutation $\pi$ of the vertex indices, the optimization objective is:

$$\pi^* = \arg\min_{\pi} \left( \sum_{i=1}^{n-1} w(v_{\pi_i}, v_{\pi_{i+1}}) + w(v_{\pi_n}, v_{\pi_1}) \right). \tag{1}$$

**Bin Packing Problem**. The BPP is another classical NP-hard problem in combinatorial optimization. An instance consists of a set of $n$ items $I = \{i_1, i_2, ..., i_n\}$ with associated sizes $S = \{s_1, s_2, ..., s_n\}$, and an infinite supply of bins, each with a fixed capacity $C$. The objective is to partition the item set $I$ into the minimum number of disjoint subsets $B_1, B_2, ..., B_k$, where each subset corresponds to the contents of a bin, subject to the capacity constraint:

$$\forall j \in 1, ..., k, \quad \sum_{i_m \in B_j} s_m \leq C. \tag{2}$$

The optimization goal is to minimize $k$, the total number of bins used. In the offline setting, all items are known in advance, while in the online setting, items arrive sequentially and must be placed before the next item is revealed. In this work, we focus on the offline setting.

## 3.2 Automatic Heuristics Design

Automatic heuristics design aims to automatically select, refine, combine or generate high-performance heuristics for a specific problem or class of problems. Its core objective is to explore a vast design space of heuristics to discover algorithms that can efficiently solve complex optimization or search problems, thereby reducing the reliance on manual design and expert knowledge. This process can be formally defined as searching within a given heuristic space $H$ to find an optimal heuristic $h^*$ that maximizes the final performance evaluated by a function $g(\cdot)$ over a specific set of problem instances $I$:

$$h^* = \arg\max_{h \in H} g(h). \tag{3}$$

The heuristics space $H$ represents the set of all candidate heuristics that can be constructed or selected. This space can be discrete (e.g., fixed algorithmic components) or continuous (e.g., parameterized functions). Besides, the performance evaluation function $g(\cdot)$ quantifies the effectiveness of a heuristic $h$. It is typically assessed by measuring solution quality, computational cost, or other relevant metrics on a benchmark set of problem instances. These symbols provide the formal foundation for evaluating heuristics' quality, guiding selection and update processes in the 4.1.

## 4 Method

The EvoPH framework is a closed-loop system for the automatic design and optimization of heuristic algorithms. As illustrated in Figure 1, EvoPH operates through an iterative cycle that integrates two complementary components: heuristics evolution and prompt evolution. The heuristics evolution module employs an island-based elites selection algorithm to refine candidate heuristics, while maintaining diversity through migration across subpopulations. The outcomes are distilled into experience, which provide structured feedback. This feedback, in turn, drives the prompt evolution module and strategy sampling module, guiding the next generation of heuristic algorithms. Together, these processes form a continuous loop of generation, evaluation and adaptation. In the following sections, we provide a detailed discussion of each sub-process.

### 4.1 Heuristics Evolution

**Heuristic Algorithm Generation**. In the generation phase, The LLM is used as a high-level semantic mutation operator to generate new candidate algorithms. Starting from the parent algorithms selected from the elite library, the LLM generates a new generation of algorithms under the guidance of carefully designed prompts.

**Experience Summarization**. After generating candidate heuristic algorithms, EvoPH evaluates them on the given problem instances. When execution produces valid solutions, corresponding performance metrics are extracted; in cases of invalid outputs, systematic analysis and reporting are conducted. Regardless of correctness, The execution results or the analytical report is distilled into structured experiential knowledge that captures effective strategies and performance characteristics. This accumulated experience is subsequently synthesized into reflective feedback, which in turn guides the next iteration of heuristic search.

**Heuristics and Experience Storage**. The EvoPH proposes an *island-based elites selection algorithm* to organize and preserve heuristic algorithms together with their experience. The core idea is to partition the global population into $N$ relatively independent subpopulations, referred to as *islands*. Each *island* independently executes a full elites selection process, maintaining its own elite archive. While the islands evolve autonomously, they are not entirely isolated; a periodic migration mechanism enables the exchange of elite individuals, thereby promoting global information sharing and cooperative co-evolution. The details of the elites selection algorithm are as follows:

- **Feature Space Definition**. To guide elites selection, we first define a multidimensional behavioral feature space for program solutions. Intuitively, this space can be viewed as a grid, where each cell in this grid corresponds to a unique combination of behavioral features (e.g., high development potential with low relative error). Each cell stores the best-performing solution found for that feature combination. Formally, for a heuristic $h \in H$, we define a mapping function $F : H \rightarrow B$ that projects $h$ into a behavioral descriptor $b \in B$. Each island $i$ maintains

an elite archive $M_i$, in which cells are indexed by descriptors $b$ and record the best heuristic $h$ currently associated with $b$.

- **Archive Update**. When a new heuristic $h_{\text{child}}$ is generated, we first evaluate its performance $g(h_{\text{child}})$ and get its descriptor $b_{\text{child}} = F(h_{\text{child}})$. The archive is = updated in the following way:

$$M_i(b_{\text{child}}) \longleftarrow \begin{cases} h_{\text{child}} & \text{if } M_i(b_{\text{child}}) = \emptyset \text{ or } g(h_{\text{child}}) \geq g(M_i(b_{\text{child}})) \\ M_i(b_{\text{child}}) & \text{otherwise} \end{cases} \quad (4)$$

where $g(h)$ denotes the performance of heuristic $h$ as defined in Section 3.2. This update rule ensures that only heuristics with equal or superior performance replace the existing elite. Through this process, each island incrementally explores its search region, while the collective archives promote both potential and solution quality in the global search.

- **Heuristic Selection for Evolution**. After updating the elite archive, EvoPH selects parent heuristics for the next generation through an experience-guided process. First, a candidate island is chosen, within the selected island, EvoPH then adaptively balances exploration and exploitation based on heuristics experience: in the exploration mode, a parent heuristic is randomly sampled to promote behavioral diversity, whereas in the exploitation mode, heuristics demonstrating consistently high quality across multiple descriptors are prioritized. This mechanism enables EvoPH to simultaneously foster innovation and leverage proven solutions, thereby preventing premature convergence to local optima.

- **Island Migration**. At predefined generational intervals, migration events occur. Selected elites from a source island are introduced into the evolutionary cycle of a target island, where they compete with local elites under the same archiving mechanism. These migrated solutions enrich the diversity of the population and strengthen cooperative co-evolution across islands.

## 4.2 PROMPT EVOLUTION

The key idea of prompt evolution is to elevate the evolutionary search from the program level to the prompt level. In this meta-evolutionary framework, as heuristics undergo optimization, the instructional prompts guiding their mutation are co-evolved concurrently. This ensures that mutation operations remain both targeted and potent, continuously adapting to the state of the search. Our proposed prompt evolution consists of two primary steps:

**Prompt Update**. The prompt update step employs a closed-loop mechanism in which adaptation is guided by experiential feedback from heuristics evolution. In each iteration, the performance of generated heuristic algorithm is recorded as experience, which, together with the initial prompts, are fed back into the LLM to guide subsequent prompt refinement. Prompts associated with effective heuristics are reinforced, while those consistently leading to poor outcomes are refined or discarded. Through this iterative process, the system autonomously learns and improves prompts.

**Strategy Sampling**. To simulate the diverse mutation patterns observed in biological evolution and to introduce greater exploration potential into the evolutionary process, this study pre-designed a variety of differentiated "evolution strategies". These strategies are modularly embedded within the prompts to guide the LLM in performing different mutation operations during the heuristics evolution process. The selection of strategies is informed by accumulated experience, in which the historical performance of previously generated heuristics is recorded. The experience serves as a reference for matching problem characteristics with suitable strategies, thereby enabling the framework to adaptively sample a strategy from the pool that is most appropriate to the current search state rather than relying on fixed or random selection. A detailed description of these strategies in the pool is provided in Appendix A.1.The final prompts submitted to the LLM are dynamically combined from the iteratively updated prompts and the evolutionary strategy sampled based on experience. The specific content of each prompt is detailed in Appendix A.3.

The prompts adaptively update based on experiential feedback, inheriting knowledge from historical successes while avoiding repeated failures. In doing so, accumulated experience guides both the refinement of prompts and the sampling of evolution strategies, enabling the system to select the most appropriate mutation pathway for the current search context. Such an experience-driven mutation mechanism effectively aids heuristic algorithm populations in escaping local optima, significantly improving algorithm discovery efficiency and solution quality. At the same time, it enhances the

effectiveness of individual mutations at the micro level and provides a solid foundation for sustained heuristics evolution at the macro level.

### 4.3 COMPARISON TO PREVIOUS WORK

EvoPH advances beyond prior methods in several key aspects. **First**, it shifts the search focus from directly evolving heuristic algorithm to evolving LLM-generated prompts, which allows for a richer and more flexible exploration of the heuristic space. In contrast, FunSearch (Romera-Paredes et al., 2024) evolves only heuristic using genetic operators. **Second**, EvoPH maintains a diverse heuristic population through an island-based elites selection mechanism and enhances it with an experience-driven adaptation loop that dynamically adjusts evolution strategies, thereby ensuring both stability and adaptability; by comparison, EoH (Liu et al., 2024a) relies on fixed prompt strategies, and ReEvo (Ye et al., 2024) employs LLM-based reflection while maintaining a population but does not evolve prompts. **Third**, while NeRM (Guo et al., 2025) jointly refines prompts and algorithms with predictor assistance, EvoPH focuses exclusively on prompt-level evolution and leverages its experience-driven loop for efficient, adaptive, and targeted strategies improvement. Overall, by combining prompt-level evolution with experience-driven adaptation, EvoPH achieves broader exploration, higher efficiency, and greater robustness than existing approaches.

## 5 EXPERIMENT

For TSP and BPP, we initialize our population using a series of classic heuristic algorithms. The specific details of each algorithm can be found in Appendix A.2. In the following sections, we provide a detailed description of the dataset composition, experimental settings, and the specific experimental components of our study.

### 5.1 DATASET CONSTRUCTION

Existing automatic heuristics design methods often rely on randomly generated, fixed-size examples for performance evaluation (Ye et al., 2024). While these methods provide a set of examples, they are insufficient to capture the diversity of structural features. Moreover, such approaches fail to account for the complexity of real-world problems, which may lead to the framework learning overfit heuristic algorithms that are only applicable to a specific, idealized dataset. To address these limitations and provide a more systematic and robust evaluation of algorithm performance, we constructed two new benchmark datasets: **TSP-Gurobi-Bench** (TGB) and **BPP-Ortools-Bench** (BOB).

**TSP-Gurobi-Bench**. The TGB is derived from the classic TSPLIB database (Reinelt, 1991). We first converted the city clusters, represented by coordinates, into distance matrices. Specifically, we constructed a complete graph of cities, where each city is connected to every other city. Then we use the Gurobi solver (Gurobi Optimization, LLC, 2022) to compute the optimal solution for each instance. To ensure feasibility and efficiency, we excluded instances that the Gurobi could not solve within a 600-second time limit, resulting in a dataset containing 58 instances with optimal solutions.

**BPP-Ortools-Bench**. For the BOB, we used randomly generated instances from BPPlib (Corvello et al., 2010) as the data source and employed Google OR-Tools (Google, 2024) as the solver to compute the corresponding optimal solutions for each bin packing problem. After considering both problem size and computational time, we selected 92 instances to form the final BOB dataset.

**Evaluation metrics**. To ensure objective and standardized performance comparisons across all algorithms, we use the *relative error* as the primary quantitative evaluation metric. The lower the relative error, the better the performance. The *relative error* is defined as follows:

$$\text{Relative Error} = \frac{A_{\text{sol}} - O_{\text{sol}}}{O_{\text{sol}}} \times 100\% \tag{5}$$

where $O_{\text{sol}}$ represents the optimal solution from Gurobi or OR-Tools, and $A_{\text{sol}}$ represents the solution obtained by the heuristic algorithm.

Table 1: The experiment result of different methods on TGB and BOB datasets. "BASE" represents the relative error of the initialized heuristic algorithm. The smaller the relative error, the better the algorithm. **Bold** fonts denote the best performance.

| Dataset | Heuristics | BASE | Funsearch | EoH | mEoH | Reevo | EvoPH |
|---------|------------|------|-----------|-----|------|-------|-------|
| **TGB** | Christofides | 20.64% | 19.71% | 9.64% | 16.90% | 20.60% | **5.17%** |
| | 2-opt | 6.62% | 6.62% | 7.00% | 6.67% | 6.58% | **4.20%** |
| | nearest-insertion | 19.54% | 19.54% | 8.78% | 11.60% | 19.50% | **4.41%** |
| | farthest-insertion | 8.20% | 7.20% | 8.00% | 8.00% | 8.20% | **4.05%** |
| | nearest-neighbor | 24.67% | 16.50% | 7.80% | 24.67% | 24.67% | **4.41%** |
| | random-insertion | 9.43% | 8.11% | 9.11% | 8.90% | 9.34% | **4.41%** |
| **BOB** | first-fit | 4.90% | 4.90% | 4.90% | 4.90% | 4.90% | **0.43%** |
| | best-fit | 28.13% | 25.49% | 17.20% | 23.45% | 26.77% | **1.65%** |
| | next-fit | 5.61% | 5.61% | 5.61% | 5.61% | 5.61% | **1.59%** |
| | worst-fit | 14.66% | 7.66% | 4.90% | 14.66% | 14.66% | **1.65%** |

## 5.2 EXPERIMENT SETUP

**Baseline**. We rigorously evaluate the proposed EvoPH framework in the TGB and BOB . The comparison group includes **Funsearch** (Romera-Paredes et al., 2024), **EoH** (Liu et al., 2024a), **mEoH** (Yao et al., 2025) and **Reevo** (Ye et al., 2024). By contrasting the performance of our framework with these methods, we aim to objectively assess its superiority and effectiveness in solving COPs.

**Implementation details**. The Gemini-2.5-pro model is used as the heuristics generation and optimization model, with its inference temperature set to 0.8 and top-$p$ set to 0.95 to ensure both diversity and logical consistency in the generated content. The remaining key hyper-parameters for the evolutionary process are as follows: the maximum number of iterations is set to 20. To promote global search and maintain population diversity, the entire population is divided into 5 independent islands for collaborative evolution. In each evolutionary generation, the newly generated candidate programs are evaluated through a standardized evaluation process, with each execution time strictly limited to a 600-second threshold.

## 5.3 EXPERIMENT RESULT

### 5.3.1 MAIN RESULT

**TSP Results**. As shown in Table 1, the EvoPH framework achieves substantial performance improvements on the TGB dataset across all six initialization heuristics. For instance, for the *Christofides* and *nearest-insertion* heuristics, EvoPH consistently outperforms competing approaches, lowering errors from 20.64% and 19.54% to 5.17% and 4.41%, respectively. Even in cases where baselines are already strong, such as *2-opt*, EvoPH achieves further gains, reducing the error from 6.62% to 4.20%. These results not only highlight EvoPH's robustness in handling both strong and weak initial heuristics but also demonstrate its excellent generalization ability across diverse algorithmic starting points.

**BPP Results**. As shown in Table 1, EvoPH achieves remarkable improvements on the BOB dataset across all initialization heuristics. For example, for the *next-fit* and *worst-fit* heuristics, which stagnate at 5.61% and 14.66% in the baselines, EvoPH lowers the errors to 1.59% and 1.65%, respectively. Most notably, EvoPH achieves a dramatic reduction for the *best-fit* heuristic, from 28.13% to 1.65%, whereas competing approaches such as Funsearch, EoH, and Reevo fail to deliver comparable improvements. These results highlight EvoPH's strong cross-domain adaptability.

### 5.3.2 ABLATION RESULTS

To investigate the individual contributions of each core component within our proposed framework, we conduct a series of detailed ablation experiments on the TGB dataset, comparing the outcomes with those obtained from the complete framework. We design the following three ablation variants: (1) *w/o Strategy Sampling*: This variant removes the strategy sampling component from the frame-

work; (2) *w/o Prompt Evolution*: This variant replaces the dynamic prompt evolution module with a fixed prompt strategy; (3)*w/o Island-Based Elites Selection*: This variant removes the island model and the elites selection algorithm from the framework.

Table 2: Performance comparison of the EvoPH framework and its different ablation versions on various heuristic algorithms. Here, SS denotes *Strategy Sampling*, PE denotes *Prompt Evolution*, and IES denotes *Island-based Elites Selection*.

|  | nearest-insertion | 2-opt | Christofides | farthest-insertion | nearest-neighbor | random-insertion |
|---|---|---|---|---|---|---|
| EvoPH | 4.41% | 4.20% | 5.17% | 4.05% | 4.41% | 4.41% |
| w/o SS | 5.17% | 4.38% | 5.17% | 5.17% | 5.17% | 4.41% |
| w/o PE | 5.17% | 5.17% | 9.24% | 5.50% | 5.17% | 5.17% |
| w/o IES | 9.70% | 5.17% | 7.03% | 5.99% | 6.48% | 6.90% |

**Results**. As shown in Table 2, the ablation results clearly demonstrate the necessity and effectiveness of each component within our framework, highlighting the significant synergy between modules. Specifically, without Strategy Sampling, performance drops across all tasks, showing that maintaining policy diversity is essential to ensure broader exploration. Without Prompt Evolution, the performance drops noticeably (e.g., Christofides from 5.17% to 9.24%), indicating that adaptive prompt updates are crucial for guiding effective mutations. Without Island-based Elites Selection, performance deteriorates significantly (e.g., nearest-insertion from 4.41% to 9.70%), confirming that the island model based elites selection act as the foundational mechanism for sustaining both robustness and high-quality solutions.

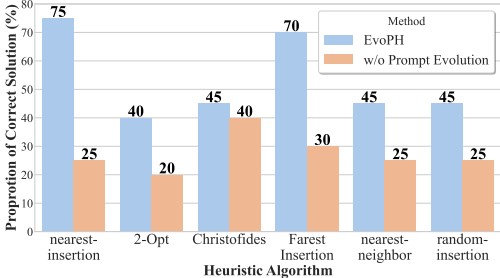

Figure 2: Proportion of generating executable code over 20 iterations of the EvoPH with and without the prompt evolution module.

Figure 3: Variation in the lowest relative error of different initial algorithms with evolution iterations.

## 5.4 ROBUSTNESS AND EFFECTIVE OF EVOPH

We conducted further experiments to assess the robustness of EvoPH in terms of both heuristics executability and convergence behavior. Under the same experimental setting, we first examined the proportion of generating executable heuristics over 20 iterations, comparing EvoPH with and without the prompt evolution module. As shown in Figure 2, incorporating prompt evolution consistently leads to higher success rates across different heuristics, substantially improving the likelihood of producing reliable code within a limited number of iterations.

In addition, we investigated the convergence behavior of multiple heuristic algorithms initialized with different algorithms. As illustrated in Figure 3, despite initial performance gaps, all algorithms follow a similar convergence trajectory: rapid quality improvements in early iterations, slower gains thereafter and eventual stabilization. This consistent trend highlights the general effectiveness of EvoPH across diverse initialization strategies.

## 🚀 (Before Evolve) Evolutionary Goal and Evolutionary Directives

Your primary objective is to enhance the provided function in each evolutionary step, pushing it towards the optimal balance of solution quality and speed. Assume you are receiving a function that already exists and needs improvement.

In each round, you must analyze the function provided to you and then rewrite it to be better.

## 🚀 (After Evolve) Evolutionary Goal and Evolutionary Directives

Your primary objective is to enhance the provided function in each evolutionary step, pushing it towards the optimal balance of solution quality

The returned code must be valid Python code that executes without errors. Prioritize generating correct and functional code above all else. If a change results in worse performance, revert to the previous version and explore alternative strategies. Critically analyze the traceback and error messages to understand the failure's context within the code.

When the code is not valid, focus exclusively on fixing the errors. Do not attempt optimizations until the code is error-free. Carefully examine the provided `error_reason` and traceback information to pinpoint the root cause. Use print statements or a debugger to understand the program's state at the point of failure. Consider edge cases and boundary conditions, especially when array slicing or manipulating indices. Test your fix thoroughly with small example inputs before submitting.

When the code is valid concentrate on improving performance, Experiment with different neighborhood search strategies, candidate lists, and data structures tailored to these algorithms. Consider the trade-offs between exploration and exploitation.

Figure 4: Comparison of prompts before and after evolution.

```
Before Evolution
def solve_tsp_approximate(dist_matrixnp.ndarray):
…
  improved=True
    while improved:
      improved=False
      for I in range(n-1):
        for j in range(i+2,n):
          a,b,c,d=tour[i-1],tour[i],tour[j-1],tour[j]
          delta=dist_matrix[a,c]+dist_matrix[b,d]
              -dist_matrix[a,b]-dist_matrix[c,d]
          if delta<0:
            tour[i:j]=tour[i:j][::1],
            improved=True,
            break
        if improved:
          break
…
```

```
After Evolution
def solve_tsp_approximate(dist_matrix: np.ndarray):
…
  improved = True
    while improved:
      improved = False,best_change = 0,best_swap = None
      for i in range(n - 1):
        for j in range(i + 2, n):
          a, b = tour[i], tour[(i + 1) % n]
          c, d = tour[j], tour[(j + 1) % n]
          change=dist_matrix[a, b]+dist_matrix[c,d]-(dist_matrix[a,c] + dist_matrix[b,d])
          if change > best_change:
            best_change = change
            best_swap = (i, j)
        if best_swap:
          i, j = best_swap
          tour[i+1:j+1] = reversed(tour[i+1:j+1])
          improved = True
…
```

Figure 5: Comparison of heuristic algorithm before and after evolution.

### 5.5 CASE STUDY

**Case of prompt evolution**. As shown in Figure 4, the evolved prompts demonstrate clearer structure and stronger task orientation than their pre-evolution counterparts. They adopt a hierarchical instruction framework that prioritizes code correctness and functionality. When the code is invalid, the prompts restrict the task to error repair, specifying detailed steps such as analyzing error messages, applying debugging tools, and verifying fixes. Once validity is ensured, the focus shifts to performance optimization with explicit strategies. This evolution enables the prompts to reliably guide the model from securing functional correctness to pursuing performance improvements.

**Case of heuristics evolution**. According to Figure 5, compared with the pre-evolutionary algorithm that adopts the first-improvement strategy, the key advantage of the evolved algorithm lies in its exhaustive evaluation of all possible exchanges, where instead of stopping at the first improving move, the algorithm scans through every candidate swap and identifies the globally optimal modification. By applying this best-improvement mechanism in each iteration, the evolved algorithm achieves a more consistent and thorough enhancement of solution quality.

## 6 CONCLUSION

This paper proposes a structured co-evolutionary framework, EvoPH, designed to efficiently solve combinatorial optimization problems through the iterative evolution of prompts and heuristic algorithms. The synergy between macro-level population management and micro-level co-evolution facilitates nested optimization, effectively avoiding local optima and enhancing the performance of algorithm design. Extensive experimental results on the TGB and BOB demonstrate that EvoPH outperforms existing evolutionary computation methods in terms of solution quality. Future research will focus on expanding this framework to a broader range of combinatorial optimization problems, with an emphasis on its application in practical algorithm design tasks.

## 7 ETHICS STATEMENT

This work studies automatic heuristic design for TSP and BPP using publicly available data from TSPLIB and BPPLIB. These data contain no human subjects, personal data, or sensitive information; hence, concerns regarding privacy, safety, or legal compliance do not apply. The study does not pose risks of discrimination or bias, and all experiments follow community standards of reproducibility, fairness, and research integrity. As no human or animal subjects are involved, IRB approval is not required, and no conflicts of interest or external sponsorship exist.

## 8 REPRODUCIBILITY STATEMENT

We have made extensive efforts to ensure reproducibility of our work. The main text describes the EvoPH framework, its key components (heuristics evolution and prompt evolution), and the experimental setup in detail (Sections 4–5). Benchmark datasets TGB and BOB are constructed from TSPLIB and BPPLIB, with the exact preprocessing procedures explained in Section 5.1. Complete algorithmic details, including initial heuristics, evolution strategies, and prompt templates, are provided in Appendix A. Hyperparameters, model settings and runtime configurations are specified in Section 5.2. Additionally, all datasets and code will be released anonymously as supplementary material to enable independent verification of results. Together, these resources ensure that the experiments and findings reported in this paper can be reliably reproduced and extended.

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

# A  APPENDIX

## A.1  EVOLUTION STRATEGY

In this section, we introduce the evolutionary strategies that guide mutation within EvoPH, ranging from parameter modification to completely rewrite. These strategies, embedded in the prompt design, balance stability with exploration, ensuring effective heuristics evolution. The complete set and their respective roles are summarized below:

- **Parameter modification**. The most conservative strategy instructs LLM to focus on identifying and fine-tuning hard-coded constants, thresholds, or hyper-parameters in the algorithm to explore the potential of existing algorithms without changing the core logic.

- **Redundancy Removal**. Focuses on algorithm optimization and efficiency, requiring LLM to analyze and remove unnecessary calculations, repeated logical judgments, or simplified code snippets to improve algorithm execution efficiency.

- **Structural modification**. A moderately exploratory strategy that guides LLM to adjust the existing code structure, such as changing the nesting of loops, replacing data structures, or adjusting the order of function calls.

- **Heuristic rewrite**. A more radical strategy requires LLM to identify a core heuristic rule or submodule in the algorithm and try to rewrite it with a completely new, functionally equivalent or better logic, aiming to achieve innovation in key steps.

- **Completely rewrite**. The most exploratory strategy, instructing LLM to completely abandon the existing implementation and write a completely new version from scratch while retaining the original algorithm intent (solving a specific problem). This strategy is used to make a disruptive attempt when the evolution has reached a serious stagnation.

## A.2  INITIAL HEURISTICS FOR EVOLUTION

### A.2.1  HEURISTIC ALGORITHM FOR TSP

This subsection introduces several representative heuristic algorithms for the TSP, ranging from simple greedy methods to more sophisticated approaches with theoretical guarantees.

- **Nearest Neighbor**. The Nearest Neighbor algorithm is a simple greedy heuristic that constructs a tour by starting at an arbitrary city and repeatedly traveling to the closest unvisited city. This process continues until every city has been visited, at which point the tour is completed by returning to the starting city.

- **Nearest Insertion**. The Nearest Insertion algorithm builds a tour incrementally by starting with a small sub-tour of two cities and progressively adding more. In each step, it identifies the unvisited city that is closest to any city already on the sub-tour and then inserts it into the position along the tour's edge that causes the smallest increase in total length. This process is repeated until all cities have been incorporated into the tour.

- **Farthest Insertion**. The Farthest Insertion algorithm also builds a tour incrementally but uses an opposite selection criterion from Nearest Insertion. It starts with a small sub-tour and, at each step, selects the unvisited city that is the farthest from any city currently in the sub-tour. It then inserts this selected city into the edge of the sub-tour that results in the least additional travel distance.

- **Random Insertion**. The Random Insertion algorithm constructs a tour by starting with a small initial sub-tour and then inserting the remaining cities one by one in a completely random order. For each randomly selected city, the algorithm evaluates all possible insertion points along the edges of the current sub-tour and places the city in the position that minimizes the increase in the tour's total length.

- **2-Opt**. The 2-Opt algorithm is an improvement heuristic designed to refine an existing tour by systematically eliminating edge crossings. It works by iteratively selecting two non-adjacent edges in the tour and checking if swapping their endpoints to reconnect the path in a different order would shorten the total distance. If a beneficial swap is found, the tour

is updated, and the process is repeated until no more length-reducing swaps are possible, resulting in a locally optimal solution.

- **Christofides Algorithm**. The Christofides algorithm is an advanced heuristic that provides a theoretical performance guarantee, ensuring the resulting tour is no more than 1.5 times the length of the optimal solution. It operates by first creating a Minimum Spanning Tree (MST) of all cities, then identifying all vertices with an odd degree and finding a minimum-weight perfect matching for them. By combining the MST and the matching, it forms an Eulerian circuit, which is then converted into a valid TSP tour by taking shortcuts to avoid revisiting cities.

### A.2.2 HEURISTIC ALGORITHM FOR BPP

This subsection presents several classical heuristic algorithms for the BPP, highlighting their strategies for item placement and trade-offs between efficiency and packing quality.

- **First Fit**. The First Fit algorithm is an intuitive online algorithm. It processes each item sequentially and places it in the first bin with enough free space. If no bins are found that can hold the item, the algorithm moves to a new, empty bin and places the item there.

- **Best Fit**. The Best Fit algorithm aims to use space most efficiently. For each item, it searches for the box that can accommodate it and has the least amount of remaining space, also known as the "most compact" box. If all existing boxes cannot accommodate it, it will open a new box. This strategy attempts to avoid leaving large, unusable fragmented spaces in the box, but because it needs to check all boxes, it is slightly slower than the first fit algorithm.

- **Next Fit**. The Next Fit algorithm is the simplest and fastest heuristic, but it's generally the least efficient. It maintains a single, "current" active chest and attempts to place the next item into it. If it fits, it does so. If not, the algorithm simply "closes" the current chest (no longer considering it) and opens a new one for the item.

- **Worst Fit**. The Worst Fit algorithm is the inverse of the best-fit strategy. For each item, it searches for the bin with the largest remaining space that can accommodate it. The goal is to preserve a large, contiguous area to accommodate potentially large items in the future. However, this strategy often performs poorly in practice because it tends to prematurely occupy multiple bins, resulting in inefficient overall packing.

## A.3 Prompt used in evolution

In this section, we present the initialization and evolution prompts that guide the EvoPH framework. Specifically, the initialization prompts for the TSP and the BPP define the evolutionary objectives, optimization metrics, and implementation directives for the respective tasks. In addition, a meta-level prompt is introduced to refine existing prompts based on execution feedback and performance indicators. These prompts, which serve as the foundation for both algorithmic evolution and reflective prompt refinement, are illustrated in Figure 6.

(a) Prompt template for iteratively evolving heuristics on the TSP.

(b) Prompt template for iteratively evolving heuristics on the BPP.

(c) Meta-prompt designed to refine and improve existing prompts based on execution feedback and performance metrics.

Figure 6: Initialization and evolution prompts used in EvoPH

