# OpenReview forum: "Experience-Guided Reflective Co-Evolution of Prompts and Heuristics for Automatic Algorithm Design"
_ICLR.cc/2026/Conference — ICLR 2026 Conference Withdrawn Submission_

### Official Review · Reviewer_dT8M · 2025-10-27

**Soundness:** 2
**Presentation:** 3
**Contribution:** 2
**Rating:** 2
**Confidence:** 4

**Summary:**

This paper proposes EvoPH (Experience-Guided Reflective Co-Evolution of Prompts and Heuristics), a new framework for automatic algorithm design to solve combinatorial optimization problems (COPs) like the Traveling Salesman Problem (TSP) and Bin Packing Problem (BPP).
Addressing limitations of traditional methods—such as over-reliance on expert knowledge, premature convergence to local optima, and inflexible fixed prompts—EvoPH enables the co-evolution of LLMs' prompts and heuristic algorithms. Also, it uses an island-based elite selection mechanism to maintain heuristic diversity, leverages historical experience to dynamically select optimization strategies, and iteratively refines prompts based on heuristic performance feedback.
Experiments on two benchmarks (TGB for TSP, BOB for BPP) show EvoPH achieves the lowest relative error against optimal solutions compared to baselines like FunSearch, EoH and ReEvo.

**Strengths:**

1. Adopts an island-based elite selection mechanism to maintain heuristic diversity and avoid premature convergence to local optima.
2. Leverages historical experience for dynamic selection of optimization strategies, enhancing the targeting of heuristic improvements.
3. Iteratively refines prompts based on heuristic performance feedback, boosting adaptation and reducing error propagation.
4. Validates the necessity of core components (prompt evolution, strategy sampling, island-based selection) through ablation studies, ensuring framework robustness.
5. Good writing and easy to understand.

**Weaknesses:**

1. Lack of novelty, because the core contribution of the paper, the idea of co-evolving prompts and heuristics, has been proposed in this ICML 2025 paper (https://openreview.net/forum?id=teUg2pMrF0, Ye H, Xu H, Yan A, et al. Large Language Model-driven Large Neighborhood Search for Large-Scale MILP Problems[C]//Forty-second International Conference on Machine Learning. 2025.)
2. Focuses only on TSP and BPP(offline only), lacking validation on a broader range of combinatorial optimization problems to prove generalizability.
3. Does not discuss the computational overhead of the co-evolution mechanism (e.g., island migration, prompt iteration) compared to simpler baseline methods.
4. Lacks investigation into the impact of key hyperparameters (e.g., number of islands, migration frequency) on overall performance and convergence speed.
5. Figure 1 is not very clear, especially the right half, which fails to clearly illustrate the evolution process of the prompt.

**Questions:**

1. This paper evolves algorithms from given heuristics (e.g., Christofides for the TSP) and compares the results with variants also evolved from the Christofides heuristic. However, previous works typically evolve algorithms either from scratch or using all seed algorithms, and then compare their outcomes with SOTA algorithms. In this regard, I believe this paper should also compare its evolved algorithms with SOTA algorithms.
2. This paper tests BPP on offline scenario only, what about the online BPP, like FunSearch and EoH did? And the dataset chosen of TSP and BPP are not following the previous work, such as FunSearch, EoH and ReEvo.
3. Only the Gemini-2.5-pro model was tested, what about other closed-source or open-source large language models(Claude, Gemini, Qwen, DeepSeek)?

---

### Official Review · Reviewer_GRBZ · 2025-10-30

**Soundness:** 3
**Presentation:** 3
**Contribution:** 2
**Rating:** 2
**Confidence:** 4

**Summary:**

This paper proposes EvoPH, a novel framework that automatically designs algorithms by co-evolving Large Language Model (LLM) prompts and heuristic algorithms to solve combinatorial optimization problems (COPs). The approach uses an island migration model and elite selection to maintain population diversity and enable prompts and heuristics to evolve together based on experiential feedback. The authors evaluate EvoPH on two classic NP-hard problems—the Traveling Salesman Problem (TSP) and Bin Packing Problem (BPP)—using standard benchmark datasets. Results show that EvoPH consistently outperforms state-of-the-art LLM-based automatic heuristic discovery baselines, reducing solution errors and generating more executable code. Ablation studies further reveal how each framework component contributes to performance.

**Strengths:**

- The paper introduces reflective co-evolution of both prompts and heuristics—an advancement over prior work that typically evolves only algorithms or uses fixed prompts.
- The framework is problem-agnostic, demonstrated on both TSP and BPP, showing broad applicability across different COPs.
- Empirical results show improvements over relevant baselines in solution quality.

**Weaknesses:**

- Empirical evaluation is restricted to two problem types (TSP and BPP) and focuses primarily on improving existing single heuristics rather than a wider variety of combinatorial problems or more challenging settings (e.g., larger-scale, noisy, or real-world instances).
- All experiments are conducted using Gemini-2.5-pro as the backbone LLM. Results may not generalize to other LLMs. No ablation is performed with alternative models (e.g., open-source LLMs or different vendor APIs).
- The paper does not report absolute computational metrics such as token consumption, model querying cost, or wall-clock time.
- While key competitors are included, the choice is limited to recent LLM-based approaches. Traditional evolutionary algorithm (EA) and hyper-heuristic (HH) baselines are not compared.

**Questions:**

- How does the computational efficiency of EvoPH scale with problem size and the number of evolution iterations? Please include wall-clock time, memory, and token usage statistics for reproducibility.
- Have the authors tested EvoPH with different LLM backbones? How sensitive is the framework to model quality?
- Could more heterogeneous benchmarks, such as SAT or graph coloring, be considered?
- Can the authors report average results over multiple random seeds and provide standard deviations for key metrics?

---

### Official Review · Reviewer_RJRE · 2025-11-01

**Soundness:** 2
**Presentation:** 2
**Contribution:** 2
**Rating:** 2
**Confidence:** 3

**Summary:**

A large number of LLM-guided automated algorithm design approaches have been proposed over the last years, and this paper is another one in this category. The authors attempt to evolve combinatorial optimisation algorithms for TSP and BinPacking. The approach uses LLMs to generate new candidate algorithms from existing ones, while simultaneously optimising the prompts used for this generation. The optimisation process uses an archive mechanism and an island model.

The proposed approach is compared empirically with similar approaches, including FunSearch.

**Strengths:**

The main strength of the paper are the seemingly strong results, relative to the existing methods (Funsearch, EoH, mEoH, and Reevo).

Another good point is that the paper makes an ablation study, showing the impact of the different components.

**Weaknesses:**

The methodology is weak. The paper contains no statistical analysis or treatment of variance. For example, the caption of Table 2 (ablation study) claims that the performance degrades significantly without Island-based elites selection. However, without information about sample size and variation, it is not possible to tell whether the degradation is just due to statistical variation. The reviewer acknowledge that it might be prohibitively expensive to run a large number of experiments using a commercial LLM. One could have used a small, open LLM. Also, the evolved heuristics themselves can be run locally and it is possible to obtain detailed information about the distribution of their performance.

The presentation has several weaknesses. The paper does not contain a full description of the algorithm in the form of unambigious pseudo-code (a figure with symbols and plain English text is insufficient). The paper instead uses space to fully define TSP and BinPacking (the definitions of these well-known algorithms can be moved to the appendix).

In equation (3), it is unclear how heuristics are evaluated. Clearly, they can be randomised, so their performance will have a distribution. How many runs of the heuristics are used for their evaluation? How do you compare heuristics with respect to the parameters of the performance distribution (e.g., perform slightly worse mean performance if the variance is lower?)

There is no discussion of the impact of the LLM chosen. How does the performance degrade if you use a smaller, open LLM? Are all methods compared using the same LLM? Across all experiments, how many queries did you make to Gemini?

The description of feature space has similarities to quality diversity methods, which should be cited.

There is no analysis of the generalisability and scalability of the heuristics evolved.

Avoid the term "evolution strategies" as it already has a very specific meaning within evolutionary computation.

**Questions:**

Please address the points described under weaknesses.

---

### Official Review · Reviewer_n61A · 2025-11-04

**Soundness:** 3
**Presentation:** 3
**Contribution:** 3
**Rating:** 2
**Confidence:** 4

**Summary:**

This paper proposes EvoPH (Experience-guided reflective co-Evolution of Prompts and Heuristics), a framework for automatic algorithm design that co-evolves LLM prompts with heuristic algorithms, guided by performance feedback. The framework employs an island-based elites selection algorithm to maintain population diversity and an experience-driven strategy sampling mechanism. The authors evaluate on the Traveling Salesman Problem (TSP) and Bin Packing Problem (BPP), creating benchmark datasets (TGB and BOB) with optimal solutions from Gurobi and OR-Tools. The paper demonstrates strong empirical results, achieving the lowest relative error compared to several baselines.

**Strengths:**

Well presented empirical results

Thorough ablation studies

Well-designed system

Good case studies

**Weaknesses:**

The authors present prompt and heuristics co-evolution, guided by performance performance feedback, as a primary novelty. Admittedly, the field is moving fast, yet, prompt-heuristic co-evolution is an established and actively developing research direction, therefore, novelty positioning needs revision.

The author's provided supplementary materials with code and artifacts. The code was clearly written by generative AI. There is not a problem using Generative AI to write code, however, it is problematic to not understand how the generative AI code - the authors presumably asked to be written -  functions. E.g. there is no evidence of "in cases of invalid outputs, systematic analysis and reporting are
conducted"

The Authors claim in section 4 "The EvoPH framework is a closed-loop system for the automatic design and optimization of heuristic algorithms". The code presented in the supplementary materials uses OpenEvolve, an open source implementation of Google DeepMind's AlphaEvolve system. The paper would provide better context by mentioning either.

The paper positions co-evolution as novel, but Wang et al. (2025) survey shows it's an established developing area
Missing citation and comparison with CALM (Huang et al., 2025), which does prompt evolution + LLM fine-tuning
While you cite ReEvo, you claim it "does not evolve prompts" - but it uses "mutation prompts", and reflection mechanisms that guide evolution, which is conceptually similar
Should cite AEL (Liu et al., 2023) which explicitly co-evolves prompts and algorithms.

**Questions:**

Given that Osol represents the optimal solution from Gurobi of OR-Tools, and Asol represents the solution obtained from the heuristic algorithm, is it fair to assume that the solution obtained by the heuristic algorithm will always be greater than the solution obtained otherwise? Note if Asol is less than Osol, then the Relative Error would be negative.

Why is the best-fit error greater than the first-fit?

Asol represents a solution, why is it always greater than the optimal solution?

Why the "ring topology" in migrating to an adjacent island instead of a random choice?

**Details Of Ethics Concerns:**

This is another example of a paper that could have been desk rejected based on supplementary material code being written by Generative AI and no mention being made in paper about base frameworks e.g. AlphaEvolve.

---

### Note · Authors · 2025-12-01

I have read and agree with the venue's withdrawal policy on behalf of myself and my co-authors.